# Impact of mental health stigma on help-seeking in the Caribbean: Systematic review

Jay-Bethenny Gallimore[1]*, Katya Gonzalez Diaz[2], Cerisse Gunasinghe[3,4], Graham Thornicroft[1,5], Tatiana Taylor Salisbury[1‡], Petra C. Gronholm[1,5‡]

1 Centre for Global Mental Health, Health Service and Population Research Department, Institute of Psychiatry, King's College London, London, United Kingdom, 2 Department of Public Health Sciences, Stockholm University, Stockholm, Sweden, 3 Department of Psychology, City University of London, London, United Kingdom, 4 Department of Psychological Medicine, Institute of Psychiatry, Psychology and Neuroscience, King's College London, London, United Kingdom, 5 Centre for Implementation Science, Health Service and Population Research Department, Institute of Psychiatry, King's College London, London, United Kingdom

‡ TTS and PCG are joint last authors on this work.
* jay-bethenny.gallimore@kcl.ac.uk

**Data Availability Statement:** All relevant data are within the manuscript and its Supporting Information files.

## Abstract

### Background

Mental health conditions often go untreated, which can lead to long-term poor emotional, social physical health and behavioural outcomes, and in some cases, suicide. Mental health-related stigma is frequently noted as a barrier to help seeking, however no previous systematic review has considered evidence from the Caribbean specifically. This systematic review aimed to address two research questions: (1) What is the impact of mental health stigma on help-seeking in the Caribbean? (2) What factors underlie the relationship between stigma and help-seeking in the Caribbean?

### Methods

A systematic search was conducted across six electronic databases (Medline, Embase, Global Health, PsychInfo, Scopus and LILACS). The search included articles published up to May 2022. Experts in the field were consulted to provide publication recommendations and references of included studies were checked. Data synthesis comprised of three components: a narrative synthesis of quantitative findings, a thematic analysis of qualitative findings, and a meta-synthesis combining these results.

### Results

The review included nine articles (reflecting eight studies) totaling 1256 participants. A conceptual model was derived from the meta-synthesis, identifying three themes in relation to mental health stigma and help-seeking in the Caribbean: (i) Making sense of mental health conditions'; (ii) Anticipated/Experienced stigma-related experiences and (iii) Individual characteristics.

**Funding:** The author(s) received no specific funding for this work.

**Competing interests:** The authors have declared that no competing interests exist.

## Conclusion

This review provides insights into the relationship between mental health stigma and help-seeking in the Caribbean based upon the current research evidence. This can be applied in the design of culturally appropriate future research, and to support policy and practice towards stigma reduction, and improved mental care help-seeking in the Caribbean.

## Introduction

Mental health conditions are one of the leading causes of disability worldwide [1]. In the Caribbean, increasing concerns over growing rates of mental health conditions have been identified [2]. There are many contributory factors to this anticipated increase in prevalence, including an increasingly aging population, and economic decline, which is linked to another major influence—the consequences of the COVID-19 pandemic [3, 4]. Traditionally in the Caribbean, poor mental health and expressing emotions has broadly been culturally and socially stigmatised, associated with shame, personal weakness, and a lack of commitment to God, which acts as a barrier to seeking mental health support [5–7].

Goffman [8] defined *stigma* as the 'situation of the individual who is disqualified from full social acceptance' (p9). Stigma can be a key contributor to the negative attitudes that surround mental health [9]. Phrases such as "mental illness", "mental health problems" and "mentally ill", can carry negative connotations and frame an individual's identity as a product of their condition. For this reason, this paper will be using the terms "mental health" and "mental health conditions", although it is understood that these phrases have been used interchangeably with other terminology.

Stigma can lead to the underutilization of mental health services, and has been identified as an important barrier to help-seeking [10]. To date, three main systematic reviews [10–12] have explored mental health-related stigma and help-seeking. All these reviews found that stigma had a clear and negative impact on help-seeking. These reviews included studies primarily from the USA, Canada, Europe or Australia and New Zealand. None of the reviews included studies in the Caribbean. This may limit the generalisations that can be drawn to this region, as a systematic review by Mascayano et al [13] identified there to be cultural nuances in mental health-related stigma in Latin America and the Caribbean, differing from those identified in studies conducted in Western European countries.

There has only been one review that has had a focus on the impact of mental health stigma on help-seeking in the Caribbean. A scoping review by Gonzalez [14] investigated the "Impact of stigma on help-seeking behaviour for people with mental disorders in Latin America and the Caribbean". There were 11 studies included, three of which were from the Caribbean. This review provided a useful preliminary assessment of the relationship between mental health-related stigma and help-seeking in these contexts, identifying stigma as a barrier to help-seeking. It is important to note that there are cultural differences between Latin America and the Caribbean, so the conclusions drawn from the Latin American studies may not be relevant to the Caribbean cultural context. There may also be differences between Caribbean countries, as this region is highly heterogenous [15]. Thus, there is scope and value in conducting an updated search and a comprehensive synthesis of the evidence in the Caribbean context specifically through a systematic review.

This systematic review is the first to explore the impact of mental health stigma on help-seeking in the Caribbean. This review aims to synthesise current evidence on the impact of

mental health stigma and help-seeking in the Caribbean. This will be addressed through two research questions:

1. What is the impact of mental health stigma on help-seeking in the Caribbean?

2. What factors underlie the relationship between mental health stigma and help-seeking in the Caribbean?

## Method

The review protocol was registered a-priori with PROSPERO (ID: CRD42022319634) and was conducted in line with the Preferred Reporting Items for Systematic Reviews and Meta-Analyses (PRISMA) guidelines (see S1 Checklist).

### Search strategy and selection of studies

Six electronic databases (Medline, Embase, Global Health, PsychInfo, Scopus and LILACS) were searched in May 2022. Subject headings and keywords were related to the Caribbean, mental health, and stigma (full list of search terms shown in S1 Appendix). Help-seeking was not included as a subject heading with relevant keywords to not limit the amount of potentially relevant studies that may be generated from this initial search. Experts in the field were contacted to provide further publication recommendations, and reference lists of included papers were hand-searched by the main author (J-BG) for eligible papers not detected in the original search.

The inclusion criteria were full-text, data-based, peer-reviewed articles, of any study design (quantitative, qualitative or mixed methods), published up to May 2022 in English or Spanish (see Table 1). Studies were eligible for inclusion if they explored the relationship between mental health-related stigma and mental health-related help-seeking in individuals living in the Caribbean. All types of stigma were included in this review, and the definitions used are guided by the Lancet Commission's report on stigma and discrimination in mental health [16]. This includes: self-stigma, stigma by association, public stigma, and structural discrimination. This review includes all areas of help-seeking, including attitudes, intentions, and behaviours.

All results from this search were imported into Endnote20 (Clarivate Analytics), where deduplication was applied. The remaining studies were uploaded to Rayyan [17] for title and abstract screening. All titles and abstracts were screened by author J-BG. Authors KGD and CG independently screened a randomly selected 20% of the sample (10% each) for consistency, and to help ensure all relevant studies were included in the review. Discrepancies were resolved by discussion and arbitration, where any further clarification over the inclusion criteria was discussed and amendments appropriately made.

Full-text papers were obtained and assessed by author J-BG against the inclusion and exclusion criteria. Authors KGD and CG independently screened a randomly selected 10% sample each. Discrepancies were resolved by discussion, and where necessary arbitration with another author (PCG). There was uncertainty over one paper, where author PCG made the decision as to whether it should be included. Author KGD assessed full papers published in Spanish for eligibility solely due to the linguistic abilities of the team.

### Data extraction

Author J-BG extracted data from all included studies using a review-specific data extraction form developed using Microsoft Excel. Data extracted from all study types included: author, year, title, aim, country, study design, sample characteristics, type of mental health condition

**Table 1. Inclusion and exclusion criteria.**

| Participants | | |
| --- | --- | --- |
| | *Include* | Individuals living in the Caribbean |
| | *Exclude* | Persons help-seeking on behalf of another individual<br>Professional caregivers<br>Caribbeans that do not live in the Caribbean region |
| **Stigma** | | |
| | *Include* | Any type of stigma relating to or associated with mental health conditions including: |
| | | • Self stigma, also known as internalized stigma (when an individual applies negative views and attitudes towards themselves) |
| | | • Stigma by association, also referred to as affiliate or courtesy stigma (experiencing disapproval or discrimination due to the association with stigmatized individuals) |
| | | • Public stigma, which includes knowledge (misinformation), attitudes (prejudice) and behaviour (discrimination) |
| | | • Structural stigma, also known as systematic or organisational stigma (inequities that result from laws, policies and practices) |
| | *Exclude* | Stigma relating to other social attributes |
| | | Stigma relating to HIV/AIDS, cancer, sexual behaviour, abortion, epilepsy, leprosy or other situations not directly focusing on mental health |
| **Help-Seeking** | | |
| | *Include* | Help-seeking for a mental health condition or any self-defined psychological, emotional or behavioural concern<br>Measures of help-seeking-related attitudes, intentions and behaviours and relating to any stage of help-seeking from seeking initial informal help to service use |
| | *Exclude* | Help-seeking for reasons other than mental health-related concerns, e.g., intellectual disabilities, epilepsy, or dementia |
| **Study Type** | | |
| | *Include* | Data-based, full-text, peer-reviewed articles |
| | | Study designs that include any type of quantitative, qualitative or mixed method studies |
| | | Articles published in English or Spanish |
| | | Articles published up to May 2022 |
| | *Exclude* | Non-data based or non-peer-reviewed articles, e.g., conference proceedings, revisions, research protocols, editorials, comments, letters, and dissertations or other 'grey literature' |

addressed, aspect of stigma explored, aspect of help-seeking explored and key findings relating to the impact of mental health-related stigma on help-seeking–extracted from both original data and author's reflections from the results section only, unless results and discussion sections were combined. For quantitative studies, the measures of stigma and help-seeking used were also extracted. For qualitative studies, the method of data collection and analysis, and relevant data extracts and authors comments were recorded. Author KGD independently extracted data from two randomly selected papers of each design type for consistency. The independently extracted results were compared, and any discrepancies were discussed and resolved. It was established that consistency in extraction had been achieved, following which data extraction for the remaining papers were conducted by author J-BG only.

## Quality appraisal

Author J-BG conducted a quality assessment of all included studies using the Mixed Methods Appraisal Tool (MMAT) [18]. Five quality criteria are listed for each study design (e.g., Is the sample representative of the target population? Are the findings adequately derived from the data?), where responses are 'Yes', 'No' and 'Can't tell'. Following the guidance from Hong et al. [18], rather than calculating an overall score for each paper, a more detailed presentation of

the ratings of each criterion was constructed to appraise the quality. Authors J-BG and KGD independently assessed the same two quantitative and qualitative papers to determine consistency, after which the remaining papers were appraised by author J-BG alone.

### Data synthesis

Data synthesis was conducted in three stages. Following guidance from the Evidence for Policy and Practice Information and Co-ordinating Centre (EPPI-Centre; [19], quantitative and qualitative research evidence were considered independently before merging.

First, a narrative synthesis [20] was conducted on findings from quantitative evidence. A preliminary synthesis was developed, providing an initial description of the results of the included studies. The relationship between mental health-related stigma and help-seeking within and between studies was explored, with continuous reference to the extracted data, to help identify factors that influence this relationship. Lastly, the robustness of the synthesis was assessed. This was achieved by appraising the methodological quality of the included studies using the MMAT, and how the evidence was synthesised [21]. A textual narrative is provided in the results.

The second stage involved a thematic analysis [22] conducted on findings from qualitative evidence. Extracted data was input into the qualitative analysis software NVivo (Release 1.7.1). Initial codes were generated to develop a codebook by author J-BG, which continued to be revised when appropriate after the coding of each paper. Codes were then clustered into groups to explore meanings, interconnections, and patterns. This led to the development of initial themes, refinement of main themes and subthemes, which were reviewed by authors PCG and TTS, to then be structured into a thematic map.

Lastly, the data underwent a process of triangulation through a thematic meta-synthesis, where the findings from the quantitative and qualitative syntheses were integrated. This involved two stages. First, the quantitative-based synthesis papers were re-reviewed to see which factors were also present in the qualitative evidence. Next, it was determined whether any relevant factors were identified in the quantitative papers that did not arise from the qualitative papers, and vice versa. The thematic map developed from the qualitative synthesis was extended to reflect findings from the quantitative synthesis.

## Results

The initial database search returned 3,765 potentially relevant papers (see Fig 1). Excluded as duplicates were 1,682 papers. A further 2,002 papers were excluded as ineligible under the review criteria. The remaining 81 papers were full-text assessed (details of the papers excluded at full-text stage are described in S2 Appendix). Nine papers met the review inclusion criteria. No additional results were obtained through contacting experts or searching through reference lists. A summary of the included papers is provided in Table 2, with full details described in S3 Appendix.

All included papers were published between 2011 and 2021, and were written in English. The number of participants ranged from three to 408 subjects. Two papers used data from the same study [23, 24]. Studies were conducted in Jamaica (37.5%, n = 3), Trinidad (12.5%, n = 1), Haiti (12.5%, n = 1), Cuba (12.5%, n = 1), Puerto Rico (12.5%, n = 1), and St Vincent and the Grenadines (12.5%, n = 1). Six (66%) papers reported quantitative data, and three papers (33%) reported qualitative data. Five papers (55%) had a main focus on both stigma and help-seeking. Three of the five papers explored public stigma and help-seeking attitudes, one focused on public stigma and help-seeking behaviours, and one on both public and cultural stigma and help-seeking behaviours. The remaining four papers (44%) focused on help-

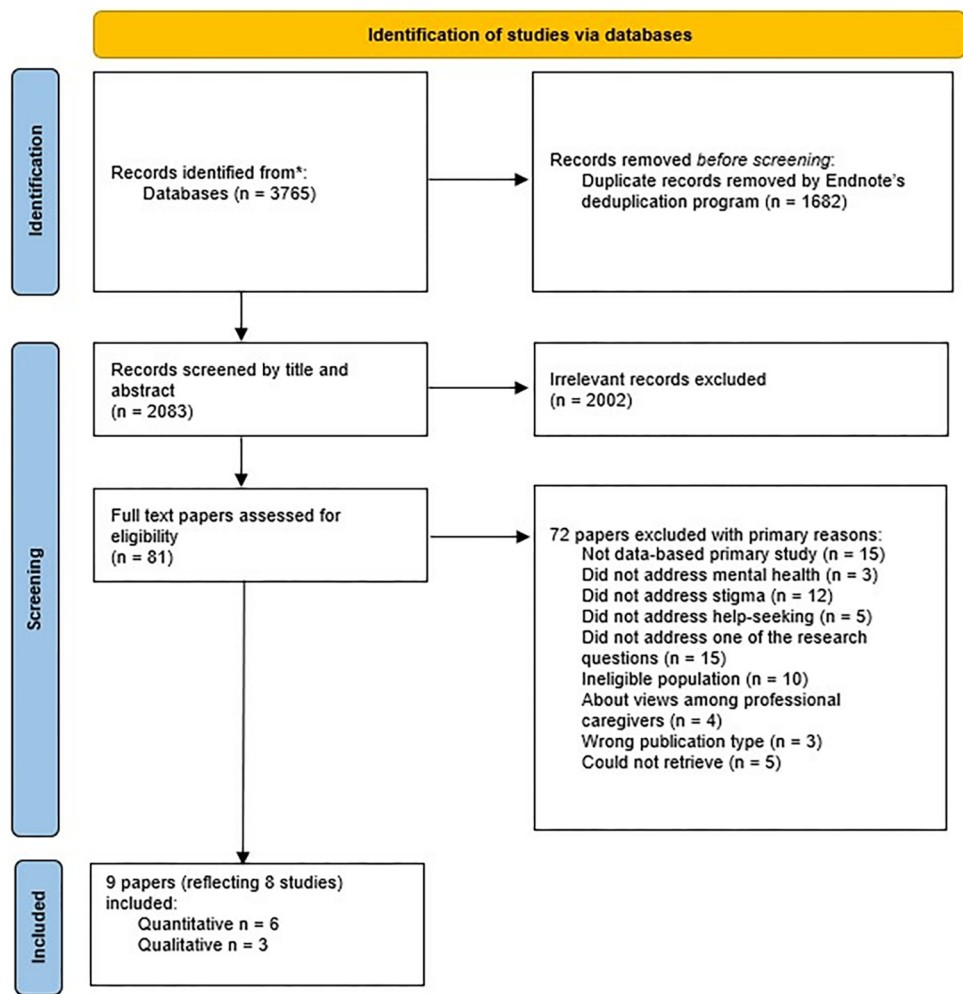

**Fig 1. Preferred Reporting Items for Systematic Reviews and Meta-Analyses (PRISMA) flow diagram.**

seeking attitudes and intentions, where stigma was an element of the study. In only one paper (11%), did all participants have experience of mental health difficulties. Four papers (44%) did not specify a mental health condition, but instead focused on mental health/mental health conditions as a whole. Specific conditions were explored across the remaining 5 articles (55%): depression, anxiety, schizophrenia, attention deficit/hyperactivity disorder-combined type (ADHD), conduct disorder, eating disorder not otherwise specified, substance abuse, bipolar disorder, and alcoholism.

The overall methodological quality of the included papers was moderate, with the quantitative papers being of poorer standard. One qualitative paper (11%) satisfied all five MMAT criteria, one (11%) satisfied four criteria, and the other (11%) satisfying three criteria, with issues relating to the appropriateness of the qualitative approach, data collection methods, and how findings were derived from the data. Three quantitative papers (33%) satisfied three out of five criteria, and the remaining three papers (33%) met two out of five criteria. This was mostly due to issues concerning sample representativeness, appropriateness of methods, and a lack of reporting on response rates. An overview of the quality appraisal of the quantitative and qualitative papers is presented in Table 3.

**Table 2. Summary of included papers.**

| Author | Country | Study Design | Sample Characteristics | Aspect of Stigma | Aspect of Help-Seeking | Key findings |
|---|---|---|---|---|---|---|
| **Quantitative Papers** | | | | | | |
| **Jackson Williams [23]** | Jamaica | Cross-sectional survey | 146 male and 193 female adolescents | Public Stigma, Perceived Stigma | Attitudes | "...for a 'psychological problem'...results indicate that students would seek help first from a medical doctor, followed by a faith healer and then from their teacher. Friends and family members were the last source of help...Across all disorders, with the exception of schizophrenia, students consistently identified friends and family members as their first choice of help. For schizophrenia, students reported that they would seek help from a psychologist/ psychiatrist first" (p467, 469) |
| **Jackson Williams [24]** | Jamaica | Cross-sectional survey | 146 male and 193 female adolescents | Public Stigma | Attitudes | "results indicate that more negative opinions about mental illness, or more authoritarian and socially restrictive opinions, as well as less benevolent opinions were associated with less positive psychological help-seeking attitudes" (p371) |
| **Maloney et al. [25]** | Jamaica | Cross-sectional survey | Survey 1–107 adolescents Survey 2–56 of the 107 adolescents from Survey 1 | Public Stigma | Attitudes | "When asked to indicate barriers to seeking mental health care, respondents most frequently reported the problem was too personal/embarrassing...or not serious enough" (p4) |
| **Nohr et al. [26]** | Cuba | Cross-sectional survey | 136 female and 59 male adults | Public Stigma | Attitudes | "community attitudes were a significant predictor of help-seeking attitudes" (p8) |
| **Ramkissoon et al. [27]** | Trinidad | Cross-sectional survey | 136 female and 22 male university students | Public Stigma | Behaviours | "Significant associations existed between perceiving mental illness to be caused by supernatural factors, seeking religious/ spiritual intervention... and seeking religious/ spiritual intervention as the first in the health-seeking pathway" (p332) |
| **Wagenaar et al. [28]** | Haiti | Cross-sectional survey | 408 adults | Public Stigma, Cultural Stigma | Intentions | "Persons who stated that suffering from mental distress is never an individual's fault were 3.5 times as likely as others to respond that they would turn to God first over hospitals or clinics and were .4 times as likely to respond that they would go to other community-based providers first compared with hospitals or clinics. Individuals responding that disasters can cause mental distress were 2.8 times as likely to respond that they would turn to God over hospitals or clinics." (p368-369) |
| **Qualitative Papers** | | | | | | |
| **Hannold et al. [29]** | Puerto Rico | Semi-structured interview | 8 veterans and 8 family members | Public Stigma | Attitudes | "Veterans may deny the need for psychological treatment because of stigma surrounding mental illness...FMs (family members) also perceived the stigma of mental illness to be real and problematic." (p385) |
| **James et al. [30]** | Jamaica | Semi-structured interview | 3 case studies (24 years old, female, paranoid schizophrenia; 45 years old, female, bipolar; 19 years old, male, schizophrenia) | Public Stigma, Cultural Stigma | Behaviours | "The effects of the supernatural as the main cause of illness was a pervasive theme throughout the interviews. This in turn influenced the treatment that the individuals requested." (p259) |

*(Continued)*

**Table 2.** (Continued)

| Author | Country | Study Design | Sample Characteristics | Aspect of Stigma | Aspect of Help-Seeking | Key findings |
|---|---|---|---|---|---|---|
| **Liu et al. [31]** | Saint Vincent and the Grenadines | Semi-structured interview | 30 church leaders | Public Stigma | Attitudes | "Those who had drinking problems tended to stray from and avoid the church, largely out of fear of condemnation: 'There are some sins members wouldn't want to confess because it's too shameful. Alcoholism is one of those issues.' " (p1086) |

## Quantitative synthesis

Six papers reported quantitative evidence, captured from 1,207 participants. All studies were cross-sectional self-completed surveys. The quality of the quantitative papers overall was average to below average, meeting two to three out of five criteria. The narrative synthesis of these data is presented in terms of: Associations between stigma and help-seeking and Stigma-related barriers to help-seeking.

**Associations between stigma and help-seeking.** Four papers reported an association between stigma and help-seeking, including data from 1100 participants. Two studies (50%) were conducted with the general population [26, 28]. The remaining two studies were conducted on adolescents [24] and university students [27].

Summarising the statistically significant findings, negative community attitudes significantly predict negative help-seeking attitudes [26]. Opinions of mental health conditions that were authoritarian, socially restrictive, or less benevolent, reflecting the attitudes aspect of public stigma, were associated with greater negative attitudes towards professional help-seeking [24]. A belief that the cause of mental health conditions is not the fault of the individual, reflecting the attitudes aspect of public stigma, led to a greater likelihood of seeking care from

**Table 3.** Overview of the quality of included quantitative and qualitative papers.

| | 1) | 2) | 3) | 4) | 5) | Number of criteria met |
|---|---|---|---|---|---|---|
| **Quantitative Papers** | | | | | | |
| Jackson Williams [23] | Y | n | n | ? | y | 2/5 |
| Jackson Williams [24] | Y | n | y | ? | y | 3/5 |
| Maloney et al. [25] | Y | n | n | n | y | 2/5 |
| Nohr et al. [26] | Y | n | y | ? | y | 3/5 |
| Ramkissoon et al. [27] | N | n | y | y | y | 3/5 |
| Wagenaar et al. [28] | Y | ? | n | ? | y | 2/5 |
| **Qualitative Papers** | | | | | | |
| Hannold et al. [29] | Y | n | y | y | y | 4/5 |
| James et al. [30] | Y | y | y | y | y | 5/5 |
| Liu et al. [31] | ? | y | ? | y | y | 3/5 |

Quantitative Criteria: 1) Is the sampling strategy relevant to address the research question? 2) Is the sample representative of the target population? 3) Are the measurements appropriate? 4) Is the risk of nonresponse bias low? 5) Is the statistical analysis appropriate to answer the research question?

Qualitative Criteria: 1) Is the qualitative approach appropriate to answer the research question? 2) Are the qualitative data collection methods adequate to address the research question? 3) Are the findings adequately derived from the data? 4) Is the interpretation of results sufficiently substantiated by data? 5) Is there coherence between qualitative data sources, collection, analysis and interpretation?

Quality Appraisal Checklist item* (Y = yes, criteria met; N = no, criteria not met;? = can't tell)

religious sources compared to hospitals or clinics [28]. Similarly, attributing the cause of mental health conditions to supernatural or medical causes, reflecting knowledge and attitudes aspects of public stigma were both associated with a willingness to seek help from both religious and medical sources [27].

**Stigma-related barriers to help-seeking.**   Two papers reported data on the stigma-related barriers to help-seeking [23, 25] including data from 446 participants. Both studies were conducted on adolescents.

Jackson Williams [23] found that the stigma attached to the type of mental health condition can influence and act as a barrier to the type of help-seeking sought. When participants were asked about a non-specified psychological condition, individuals were least likely to seek support from friends/family. Friends/family were considered an initial source of help for all specified mental health conditions apart from schizophrenia, where a psychologist or psychiatrist was preferred. Faith healers, teachers, and guidance counsellors were regarded as a last option for specified psychological conditions. Maloney et al. [25] reported that finding mental health conditions 'too personal/embarrassing' or 'not serious enough' also posed a barrier to help-seeking.

## Qualitative synthesis

Three articles reported qualitative evidence, consisting of 49 participants. One article (33%) considered the experiences of military veterans and their families [29]. Another paper (33%) explored individuals' beliefs and use of the supernatural in their own lived psychiatric experiences [30]. The final article (33%) considered church leaders' views [31]. The quality of the qualitative studies overall was good, with an average of four out of five criteria met.

Three themes were identified relating to the impact of mental health-related stigma and help-seeking in the Caribbean: (i) Making sense of mental health conditions; (ii) Anticipated/ Experienced stigma-related experiences; and (iii) Individual characteristics. These themes are described below alongside select illustrative quotations. Further participant quotations are provided in S4 Appendix.

**Making sense of mental health conditions.**   The first theme 'making sense of mental health conditions' has a focus on the language and meaning individuals attached to mental health—their own and/or others, and its related conditions. Three subthemes were identified in this process covering (a) labelling; (b) sociocultural factors; and (c) lack of recognition/ denial of one's own condition.

*Labelling.* This subtheme addresses the negative language attached to how individuals with mental health conditions in the Caribbean are perceived and understood. Such individuals are commonly labelled as *'crazy'* [29, 30]. It is also implied that having a condition represents a permanent component of a person's identity, *'once a psychotic, always a psychotic'* [30].

*Sociocultural factors.* Sociocultural factors, inclusive of cultural norms, religion and the belief of the supernatural, explored the impact of individual's sociocultural beliefs and values on their attitudes and perceptions of mental health. For example, the use of the bible and belief in 'witchcraft' was highlighted in understanding mental health conditions, and individuals expressed they *'prayed to God to please give me back my sanity'* [30] as a source of help-seeking.

*Lack of recognition/denial of one's own condition.* Individuals were identified as denying or not recognising their mental health needs due to stigma-related factors. *'Sometimes there are people that are so closed up that they don't want. . .to say anything' or 'don't see it at first glance'* [29]. This can lead to a reluctance to seek help, and an inability to recognise a need for mental health support.

**Anticipated/experienced stigma-related experiences.**   The second theme, 'anticipated/ experienced stigma-related experiences' captured responses that were outcomes of the unfair

treatment, prejudice, and negative connotations surrounding mental health conditions, which acted as a barrier to help-seeking. This was showcased in three subthemes: (a) social judgement, (b) discrimination, and (c) a lack of understanding.

*Social judgement.* This subtheme was showcased by the fear of, or experienced judgement/ condemnation. The anticipation of negative opinions of others was highlighted: *'came a Veteran and look—and came back crazy'* [29]. Additionally, there was a perception that mental health conditions may be identified as a 'sin' and be classed as 'shameful'.

*Discrimination.* Discrimination described the unjust bias and social rejection, including from employment and family, that can lead to exclusion by experiencing a mental health condition. Fear of discrimination was demonstrated in relation to employment, *'I want to get some job. I can't say that I am crazy. . .'* [29]. It was also expressed that those with a mental health condition, in this case substance abuse, would *'lose family, they lose everything'* [31].

*Lack of understanding.* A lack of understanding of an individual's own perception and beliefs about their mental health status can act as a barrier to help-seeking and lead to resistance and disagreements with care providers. Participants belief in *'bless[ing] their house and get[ting] rid of negative energy'* [30] was not understood and interpreted by doctors as hallucinating. Additionally, the involvement of the *'spiritual aspect of life'* [30] in their experiences was felt to be disregarded by doctors.

**Individual characteristics.** The third theme 'individual characteristics' highlights how certain characteristics of a population that can influence the stigma experienced and how this impacts help-seeking. One characteristic in particular was noted–military personnel. The stigma surrounding mental health in military personnel was highlighted, where veterans could be seen as they *'came back crazy from the army'* [29]. Consequently, veterans reported to *'create their own support groups'* [29] for help-seeking.

### Overall meta-synthesis

The results from the quantitative and qualitative syntheses have been triangulated to produce an overall meta-synthesis. A conceptual model illustrating this is shown in Fig 2. Five of the six subthemes identified in the qualitative synthesis were also captured in the quantitative data, with the exception of the subtheme 'lack of understanding'. One factor identified in the quantitative synthesis, but not in the qualitative synthesis, was the type of mental health condition and help-seeking source.

### Discussion

This is the first systematic review to examine the impact of mental health stigma on help-seeking in the Caribbean. It illustrates a comprehensive overview of the existing evidence in this research area, from both quantitative and qualitative perspectives, and the potential factors that underlie this relationship.

Overall, the narrative synthesis of quantitative studies indicate that mental health stigma was negatively associated with help-seeking, with the caveat that this association varies depending on the type of mental health condition in question and types of help-seeking sources. The thematic analysis of qualitative evidence identified three themes in relation to mental health stigma impacting help-seeking: (i) Making sense of mental health conditions; (ii) Anticipated/Experienced stigma-related experiences; and (iii) Individual characteristics. The conceptual model, derived from the meta-synthesis, visually demonstrates the multifaceted relationship between mental health stigma and help-seeking. The appraisal of the methodological quality of included studies suggests a particular scarcity of existing high-quality quantitative studies.

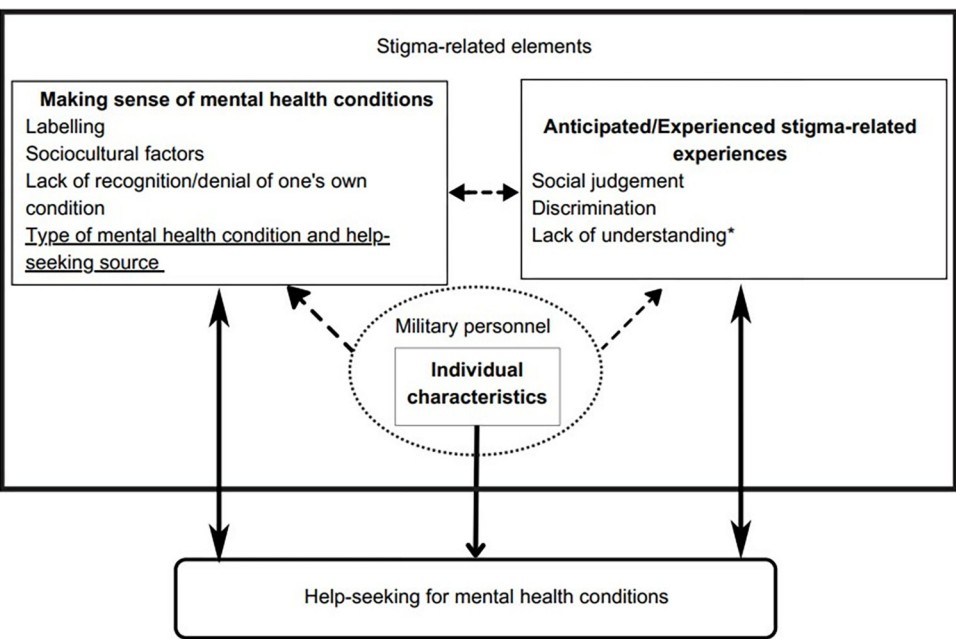

**Fig 2. Conceptual model based on meta-synthesis of qualitative (*n* = 3) and quantitative (*n* = 6) results.** Key: Boxes with solid lines represent themes and subthemes. Subthemes that are underlined were identified in quantitative studies only. Subthemes that are asterisked were reported in qualitative studies only. Subthemes that are neither asterisked or underlined were reported in qualitative and quantitative studies. Dashed arrows indicate connections between the themes. Factors within a dotted circle were identified as characteristics of a theme.

The individual quantitative and qualitative syntheses allowed for a meta-synthesis to be conducted which aided in addressing the aim of the study, to synthesise current evidence on the impact of mental health stigma and help-seeking in the Caribbean. There were only nine papers, derived from eight studies. and help-seeking, with the remaining four papers exploring stigma as a secondary point of interest. This contributed to the identification of gaps in existing evidence where further research is needed, which is discussed further in the discussion.

When exploring the subthemes that were identified in the qualitative synthesis, only one of the six subthemes was not identified in the quantitative data (lack of understanding). Whilst no subthemes were identified for the 'Individual characteristics' theme, 'military personnel' was highlighted as a characteristic from the qualitative synthesis alone. A factor from the quantitative synthesis not identified in the qualitative data was type of mental health condition and help-seeking source. This lack of overlap in findings may be a culmination of the small number of studies included, diverse populations explored, and the difference in the nature of research questions addressed in quantitative and qualitative research. The lack of mixed methods research (MMR) evidence in this review means a broader range of insights could not be captured.

This review also identified contradictory findings between studies. Jackson Williams [23] found that Jamaican adolescents did not believe seeking help would benefit mental health symptoms, whereas Maloney et al. [25] reported that most Jamaican adolescents thought digital mental health services would provide relief to those with mental health symptoms. The latter study was conducted 8 years after the former, which may suggest that young people's attitudes towards help-seeking are changing. These findings also may infer that young people prefer to engage with e-health mental health interventions, although research on this area currently has contradictory findings. Young people often look for online mental health resources,

a process that is influenced by an individual's mental health literacy [32], reflecting the knowledge aspect of public stigma, demonstrating the impact of stigma on help-seeking. Additionally, the anonymity that e-health services can provide can reduce the risk of social stigma [33], supporting a two-way relationship between stigma and help-seeking, as illustrated in this review's meta-synthesis.

Some of the factors identified in this review were only found in one study ('Lack of recognition/denial of one's own condition', 'lack of understanding', and 'military personnel' characteristic), however they remain key areas in the wider literature [11, 34–36] It is also important to note the role of 'age' in the interplay between how mental health stigma impacts help-seeking. This role has not been made explicit in the included studies to report on this, however age was explored and found to impact help-seeking. There has been support that age influences the relationship between stigma and help-seeking [37], although this role remains vague with conflicting findings in the wider literature, suggesting a need for this to be explored in the Caribbean context.

### Future research

The review identified gaps in the existing evidence and highlights future research needs. There is a dearth of research into this important topic in the Caribbean and a need for more evidence. The majority of studies in this review only focused on public stigma. Previous research has demonstrated that different types of stigma, such as self-stigma have a differential impact on help-seeking [38], thus there is a need for this to be further explored in the Caribbean context. As most research has looked at mental health as a whole, we need to know more about how stigma for different types of mental health conditions impacts different types of help-seeking. MMR would also allow for a richer understanding of this topic. Research is needed that targets the heterogeneous racial and ethnic groups that exist in the Caribbean, as well as groups under-represented in the literature and likely to be particularly impacted by stigma on help-seeking, such as military veterans and LGBTQ+ individuals. Higher quality research is also required, which will assist in overcoming some of the current methodological issues identified from the quality appraisal, such as recruiting representative samples, and using appropriate methodological measures and data collection methods.

### Implications for policy and practice

This review can be utilised in the development of policy and practice. There is a need for interventions that involve community-based mental health education and health promotion to contribute to public awareness on mental health, stigma and help-seeking. This may help to challenge and change existing labelling and negative language, as well as prevent social judgement and discrimination. When taking into account the factors and themes identified in the meta-synthesis, this indicates a need for particular attention to the needs of different groups of individuals, e.g., military personnel [29], when considering help-seeking preferences, needs and barriers, and providing support to address these. For practitioners providing treatment, there is an essential need for cultural understanding to be able to optimally engage and support an individual with a mental health condition. The results also suggest a need for self-coping strategies and teachings of ways to manage anticipated and/or experienced stigma to lessen the risk of this being a barrier to help-seeking.

### Limitations and strengths

This review restricted its selection criteria to including published, peer-reviewed papers, which may have excluded relevant papers. However, this decision was made to be able to uphold the

quality of the review. Some of the included studies had small samples, thereby caution may be needed when interpreting the findings of this review. Nonetheless, the findings from studies with lower participant numbers did not drive any single conclusions drawn in this research but rather corroborated findings from other studies, and by including all eligible studies regardless of sample size this review was able to analyse and synthesise important and interesting data from the limited existing literature to provide a thorough evaluation. Most of the included papers had many methodological issues, and there were no MMR studies, which could provide richer insights. The small number of included papers, limited Caribbean countries where the studies were set, and various target groups, restricts the generalisability of the results to other Caribbean settings. Whilst Caribbean countries share similarities, the Caribbean region is heterogenous. However, the findings from this review provide a comprehensive overview of the currently available evidence, for which research can be built upon in different Caribbean countries and contexts. The narrative synthesis, thematic analysis, and meta-synthesis were all conducted by author J-BG, which may raise issues of bias. However, analyses were reviewed by authors PCG and TTS to support the review's validity.

## Conclusion

This systematic review provides a comprehensive overview of the existing quantitative and qualitative evidence on the impact of mental health stigma on help-seeking in the Caribbean. The conceptual model that was developed from the results syntheses can help to inform future research and provide useful insight for policy and practice to prevent mental health stigma and subsequently reduce the barrier this can serve to help-seeking.

## Supporting information

**S1 Checklist. PRISMA checklist.**
(DOCX)

**S1 Appendix. Search strategies.**
(DOCX)

**S2 Appendix. Excluded papers.**
(DOCX)

**S3 Appendix. Included papers.**
(DOCX)

**S4 Appendix. Themes and subthemes with example participant quotations for included qualitative studies.**
(DOCX)

## Acknowledgments

We would like to thank King's College London Library Services for their guidance in the development of the search strategy.

## Author Contributions

**Conceptualization:** Jay-Bethenny Gallimore, Tatiana Taylor Salisbury, Petra C. Gronholm.

**Data curation:** Jay-Bethenny Gallimore.

**Formal analysis:** Jay-Bethenny Gallimore.

**Investigation:** Jay-Bethenny Gallimore.

**Methodology:** Jay-Bethenny Gallimore, Tatiana Taylor Salisbury, Petra C. Gronholm.

**Project administration:** Jay-Bethenny Gallimore.

**Supervision:** Tatiana Taylor Salisbury, Petra C. Gronholm.

**Validation:** Katya Gonzalez Diaz, Cerisse Gunasinghe, Tatiana Taylor Salisbury, Petra C. Gronholm.

**Visualization:** Jay-Bethenny Gallimore.

**Writing – original draft:** Jay-Bethenny Gallimore.

**Writing – review & editing:** Katya Gonzalez Diaz, Cerisse Gunasinghe, Graham Thornicroft, Tatiana Taylor Salisbury, Petra C. Gronholm.

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
