## [Editor Report · Decision Letter 0]

12 Jun 2023

PONE-D-23-14287Impact of mental health stigma on help-seeking in the Caribbean: systematic reviewPLOS ONE

Dear Dr. Gallimore,

Thank you for submitting your manuscript to PLOS ONE. After careful consideration, we feel that it has merit but does not fully meet PLOS ONE’s publication criteria as it currently stands. Therefore, we invite you to submit a revised version of the manuscript that addresses the points raised during the review process.

We look forward to receiving your revised manuscript.

Kind regards,

Kufre Joseph Okop

Academic Editor

PLOS ONE

Additional Editor Comments:

Abstract

Conclusion:

“This can be applied in the design of culturally appropriate future research, policy, and practice, to target and decrease stigma, and increase help-seeking in the Caribbean.”

This statement should be modified. As it is implies that the authors said this can be used to design future research and also design policy, and practice.

My suggestion:

“This can be applied in the design of culturally appropriate future research, and to support policy and practice towards stigma reduction, and improved mental care help-seeking in the Caribbean.”

Methods:

According to the authors: “This systematic review addresses two research questions viz:

(1) What is the impact of mental health stigma on help-seeking in the Caribbean?

(2) What factors underlie the relationship between stigma and help-seeking in the Caribbean?

To achieve objectives (1) and (2), there is the need to undertake a process of triangulation of the data. I have read through the data extraction and appraisal, but I have not seen any mention of triangulation. Does it mean this was captured/presented in some other terms/terminologies?

I know that Triangulation facilitates validation of data through cross verification from more than two sources. Importantly, it also tests the consistency of findings obtained through different instruments and increases the chance to control, or at least assess some of the threats or multiple causes influencing our results. Triangulation will add great value to this paper.

Results

Studies with 3 participants (mentioned in the results section) should be excluded. It is not scientifically appropriate to use findings from 3 to 5 participants (even from qualitative)

I perceive that to arrive at an efficient conceptual model, it should be based on triangulation of the data from the meta-synthesis.

Discussion, future research:

These sections are well written.

Implications for policy:

The authors said, “There is a need to incorporate mental health knowledge into early education and public awareness that includes stigma-reducing strategies”.

It seems not too logical connecting early education and public awareness to stigma reduction. I suggest that it would be more appropriate if we think of the interventions that involves community-based mental health education and health promotion that leverages public awareness on mental health, stigma and help-seeking.
---

## [Author Response · Author response to Decision Letter 0]

28 Jun 2023

Additional Editor Comments

Abstract-Conclusion:

“This can be applied in the design of culturally appropriate future research, policy, and practice, to target and decrease stigma, and increase help-seeking in the Caribbean.”

This statement should be modified. As it is implies that the authors said this can be used to design future research and also design policy, and practice.

My suggestion:

“This can be applied in the design of culturally appropriate future research, and to support policy and practice towards stigma reduction, and improved mental care help-seeking in the Caribbean.”

AUTHORS’ RESPONSE: We thank the editor for this suggestion. We have incorporated this edit into the manuscript as shown in the Abstract-page 3 lines 46-48. 

Methods:

According to the authors: “This systematic review addresses two research questions viz:

(1) What is the impact of mental health stigma on help-seeking in the Caribbean?

(2) What factors underlie the relationship between stigma and help-seeking in the Caribbean?

To achieve objectives (1) and (2), there is the need to undertake a process of triangulation of the data. I have read through the data extraction and appraisal, but I have not seen any mention of triangulation. Does it mean this was captured/presented in some other terms/terminologies?

I know that Triangulation facilitates validation of data through cross verification from more than two sources. Importantly, it also tests the consistency of findings obtained through different instruments and increases the chance to control, or at least assess some of the threats or multiple causes influencing our results. Triangulation will add great value to this paper.

AUTHORS’ RESPONSE: Thank you for raising this concern. To achieve objectives (1) and (2), a process of triangulation took place though a meta-synthesis of the quantitative and qualitative data, however we did not use the term triangulation in the original submission and express it in this way. To make it clearer for the reader, we have amended this sentence in the Method-Data synthesis section to make it explicit:

“Lastly, the data underwent a process of triangulation through a thematic meta-synthesis, where the findings from the quantitative and qualitative syntheses were integrated.” (page 9, lines 170-171).

Results - Studies with 3 participants (mentioned in the results section) should be excluded. It is not scientifically appropriate to use findings from 3 to 5 participants (even from qualitative)

AUTHORS’ RESPONSE: We thank the editor for their thoughtful response on this. As we are consolidating insights from an area where there is limited information from this geographical region, we believe it is important to include what literature is available to us. We acknowledge that studies with fewer participants may not be representative, however these studies have undergone a quality assessment, and we believe the findings should still be considered as such studies can provide rich, interesting data. 

Additionally, the findings of this study aligned with the findings of the other qualitative studies included in this review where the data contributed to common themes generated from the thematic analysis (Making sense of mental health conditions’, ‘Anticipated/Experienced mental health experiences - illustrated in Supplementary Information - S4 Appendix Themes and subthemes with example participant quotations for included qualitative studies), and thus were not making a singular point. 

We want to acknowledge in the manuscript that we recognise some of the included studies have small samples and the implications this may have, and have subsequently added the following extract into the Discussion-Limitations and strengths:

“Some of the included studies had small samples, thereby caution may be needed when interpreting the findings of this review. Nonetheless, the findings from studies with lower participant numbers did not drive any single conclusions drawn in this research but rather corroborated findings from other studies, and by including all eligible studies regardless of sample size this review was able to analyse and synthesise important and interesting data from the limited existing literature to provide a thorough evaluation.” (page 22 lines 406-411)

Results - I perceive that to arrive at an efficient conceptual model, it should be based on triangulation of the data from the meta-synthesis.

AUTHORS’ RESPONSE: Thank you for this comment. We agree that triangulation is required for an efficient conceptual model. The meta-synthesis is a result of triangulating the quantitative and qualitative data, however as we mention in a comment above, we previously did not explicitly state this. This how now been amended as demonstrated in the following sentence:

“The results from the quantitative and qualitative syntheses have been triangulated to produce an overall meta-synthesis.” (page 18, lines 313-314)

Discussion, future research - These sections are well written.

AUTHORS’ RESPONSE: We thank the editor for their positive feedback on these sections.

Implications for policy:

The authors said, “There is a need to incorporate mental health knowledge into early education and public awareness that includes stigma-reducing strategies”.

It seems not too logical connecting early education and public awareness to stigma reduction. I suggest that it would be more appropriate if we think of the interventions that involves community-based mental health education and health promotion that leverages public awareness on mental health, stigma and help-seeking.

AUTHORS’ RESPONSE: Thank you for this comment. We agree with the editor’s reflection and suggestion and have made the following edit: 

‘There is a need for interventions that involve community-based mental health education and health promotion to contribute to public awareness on mental health, stigma and help-seeking." (page 21, lines 390-392)

---

## [Decision Letter · Decision Letter 1]

29 Aug 2023

Impact of mental health stigma on help-seeking in the Caribbean: systematic review

PONE-D-23-14287R1

Dear Dr. Gallimore,

We’re pleased to inform you that your manuscript has been judged scientifically suitable for publication and will be formally accepted for publication once it meets all outstanding technical requirements.

Kind regards,

Kufre Joseph Okop

Academic Editor

PLOS ONE

Additional Editor Comments (optional):

The authors have attended to all the queries and comments - in detail. This paper is now excellently presented, and should be published. The authors are appreciated for the rigour added to their research, and for the findings/methodology that have add to science in this field of research.

Reviewers' comments:

Reviewer's Responses to Questions

**Comments to the Author**

1. If the authors have adequately addressed your comments raised in a previous round of review and you feel that this manuscript is now acceptable for publication, you may indicate that here to bypass the “Comments to the Author” section, enter your conflict of interest statement in the “Confidential to Editor” section, and submit your "Accept" recommendation.

Reviewer #1: All comments have been addressed

2. Is the manuscript technically sound, and do the data support the conclusions?

Reviewer #1: Yes

3. Has the statistical analysis been performed appropriately and rigorously? 

Reviewer #1: Yes

4. Have the authors made all data underlying the findings in their manuscript fully available?

Reviewer #1: Yes

5. Is the manuscript presented in an intelligible fashion and written in standard English?

Reviewer #1: Yes

6. Review Comments to the Author

Reviewer #1: The authors of the manuscript demonstrated mastery of the systematic review process. As a reviewer of the revised manuscript, the authors have made the necessary corrections and responded adequately to all the issues raised.

The two main questions addressed by the review was clearly stated.

The triangulation process underwent, through the thematic meta synthesis has been integrated.

Although it seemed logical and scientific for data with only 3 participants to be excluded from the review as suggested, the author's justification is profound and acceptable especially since it was reported in a qualitative research article and is a form of information in an area with dearth of research.

Including the need for interventions that involve community-based mental health education and health promotion for increased public awareness towards improving stigma reduction and help-seeking with regards to mental health is very appropriate policy implication.

The discussion, limitation and future research are well written.

7. PLOS authors have the option to publish the peer review history of their article (what does this mean?). If published, this will include your full peer review and any attached files.

Reviewer #1: No

---

## [Editor Report · Acceptance letter]

4 Sep 2023

PONE-D-23-14287R1 

Impact of mental health stigma on help-seeking in the Caribbean: systematic review 

Dear Dr. Gallimore:

I'm pleased to inform you that your manuscript has been deemed suitable for publication in PLOS ONE. Congratulations! Your manuscript is now with our production department. 

Kind regards, 

on behalf of

Dr. Kufre Joseph Okop 

Academic Editor

PLOS ONE